# TabDDPM: Modelling Tabular Data with Diffusion Models

## Abstract

Denoising diffusion probabilistic models are currently becoming the leading paradigm of generative modeling for many important data modalities. Being the most prevalent in the computer vision community, diffusion models have also recently gained some attention in other domains, including speech, NLP, and graph-like data. In this work, we investigate if the framework of diffusion models can be advantageous for general tabular problems, where datapoints are typically represented by vectors of heterogeneous features. The inherent heterogeneity of tabular data makes it quite challenging for accurate modeling, since the individual features can be of completely different nature, i.e., some of them can be continuous and some of them can be discrete. To address such data types, we introduce TabDDPM — a diffusion model that can be universally applied to any tabular dataset and handles any type of feature. We extensively evaluate TabDDPM on a wide set of benchmarks and demonstrate its superiority over existing GAN/VAE alternatives, which is consistent with the advantage of diffusion models in other fields. Additionally, we show that TabDDPM is eligible for privacy-oriented setups, where the original datapoints cannot be publicly shared. The source code of TabDDPM and our experiments is available at
https://anonymous.4open.science/r/tab-ddpm-2483.

## 1 Introduction

Denoising diffusion probabilistic models (DDPM) (Sohl-Dickstein et al., 2015; Ho et al., 2020) have recently become an object of great research interest in the generative modelling community since they often outperform the alternative approaches both in terms of the realism of individual samples and their diversity (Dhariwal & Nichol, 2021). The most impressive successes of DDPM were demonstrated in the domain of natural images (Dhariwal & Nichol, 2021; Saharia et al., 2022; Rombach et al., 2022), where the advantages of diffusion models are successfully exploited in applications, such as colorization (Song et al., 2021), inpainting (Song et al., 2021), segmentation Baranchuk et al. (2021), super-resolution (Saharia et al., 2021; Li et al., 2021), semantic editing (Meng et al., 2021) and others. Beyond computer vision, the DDPM framework is also investigated in other fields, such as NLP (Austin et al., 2021; Li et al., 2022), waveform signal processing (Kong et al., 2020; Chen et al., 2020), molecular graphs (Jing et al., 2022; Hoogeboom et al., 2022), time series (Tashiro et al., 2021), testifying the universality of diffusion models across a wide range of problems.

The aim of our work is to understand if the universality of DDPM can be extended to the case of general tabular problems, which are ubiquitous in various industrial applications that include data described by a set of heterogeneous features. For many such applications, the demand for high-quality generative models is especially acute because of the modern privacy regulations, like GDPR, which prevent publishing real user data, while the synthetic data produced by generative models can be shared. Training a high-quality model of tabular data, however, can be more challenging compared to computer vision or NLP due to the heterogeneity of individual features and relatively small sizes of typical tabular datasets. In our paper, we show that despite these two intricacies, the diffusion models can successfully approximate typical distributions of tabular data, leading to state-of-the-art performance on most of the benchmarks.

In more detail, the main contributions of our work are the following:

1. We introduce TabDDPM — the simplest design of DDPM for tabular problems that can be applied to any tabular task and can work with mixed data, which includes both numerical and categorical features.

2. We demonstrate that TabDDPM outperforms the alternative approaches designed for tabular data, including GAN-based and VAE-based models from the literature, and illustrate the sources of this advantage for several datasets.

3. We show that data produced by TabDDPM appears to be a "sweet spot" for privacy-concerned scenarios when synthetics are used to substitute the real user data that cannot be shared.

The source code of TabDDPM is publicly available [1].

## 2 RELATED WORK

**Diffusion models** (Sohl-Dickstein et al., 2015; Ho et al., 2020) are a paradigm of generative modelling that aims to approximate the target distribution by the endpoint of the Markov chain, which starts from a given parametric distribution, typically a standard Gaussian. Each Markov step is performed by a deep neural network that effectively learns to invert the diffusion process with a known Gaussian kernel. Ho et al. demonstrated the equivalence of diffusion models and score matching (Song & Ermon, 2019; 2020), showing them to be two different perspectives on the gradual conversion of a simple known distribution into a target distribution via the iterative denoising process. Several recent works (Nichol, 2021; Dhariwal & Nichol, 2021) have developed more powerful model architectures as well as different advanced learning protocols, which led to the "victory" of DDPM over GANs in terms of generative quality and diversity in the computer vision field. In our work, we demonstrate that one can also successfully use diffusion models for tabular problems.

**Generative models for tabular problems** are currently an active research direction in the machine learning community (Xu et al., 2019; Engelmann & Lessmann, 2021; Jordon et al., 2018; Fan et al., 2020; Torfi et al., 2022; Zhao et al., 2021; Kim et al., 2021; Zhang et al., 2021; Nock & Guillame-Bert, 2022; Wen et al., 2022) since high-quality synthetic data is of large demand for many tabular tasks. First, the tabular datasets are often limited in size, unlike in vision or NLP problems, for which huge "extra" data is available on the Internet. Second, proper synthetic datasets do not contain actual user data, therefore they are not subject to GDPR-like regulations and can be publicly shared without violation of anonymity. The recent works have developed a large number of models, including tabular VAEs (Xu et al., 2019) and GAN-based approaches (Xu et al., 2019; Engelmann & Lessmann, 2021; Jordon et al., 2018; Fan et al., 2020; Torfi et al., 2022; Zhao et al., 2021; Kim et al., 2021; Zhang et al., 2021; Nock & Guillame-Bert, 2022; Wen et al., 2022). By extensive evaluations on a large number of public benchmarks, we show that our TabDDPM model surpasses the existing alternatives, often by a large margin.

**"Shallow" synthetics generation.** Unlike unstructured images or natural texts, tabular data is typically structured, i.e., the individual features are often interpretable and it is not clear if their modelling requires several layers of "deep" architectures. Therefore, the simple interpolation techniques, like SMOTE (Chawla et al., 2002) (originally proposed to address class-imbalance) can serve as simple and powerful solutions as demonstrated in (Camino et al., 2020), where SMOTE is shown to outperform tabular GANs for minor class oversampling. In the experiments, we demonstrate the advantage of synthetics produced by TabDDPM over synthetics produced by interpolation techniques from the privacy-preserving perspective.

## 3 BACKGROUND

**Diffusion models** (Sohl-Dickstein et al., 2015; Ho et al., 2020) are likelihood-based generative models that handle the data through forward and reverse Markov processes. The forward process $q\left(x_{1:T}|x_0\right) = \prod_{t=1}^{T} q\left(x_t|x_{t-1}\right)$ gradually adds noise to an initial sample $x_0$ from the data distribution $q\left(x_0\right)$ sampling noise from the predefined distributions $q\left(x_t|x_{t-1}\right)$ with variances $\{\beta_1, ..., \beta_T\}$.

---

[1]URL

The reverse diffusion process $p\left(x_{0:T}\right)=\prod_{t=1}^{T}p\left(x_{t-1}|x_t\right)$ gradually denoises a latent variable $x_T{\sim}q\left(x_T\right)$ and allows generating new data samples from $q(x_0)$. Distributions $p\left(x_{t-1}|x_t\right)$ are usually unknown and approximated by a neural network with parameters $\theta$. These parameters are learned from the data by optimizing a variational lower bound:

$$\log q\left(x_0\right) \geq \mathbb{E}_{q(x_0)}\Big[\underbrace{\log p_\theta\left(x_0|x_1\right)}_{L_0} - \underbrace{KL\left(q\left(x_T|x_0\right)|q\left(x_T\right)\right)}_{L_T} - \sum_{t=2}^{T}\underbrace{KL\left(q\left(x_{t-1}|x_t,x_0\right)|p_\theta\left(x_{t-1}|x_t\right)\right)}_{L_t}\Big]$$
(1)

**Gaussian diffusion models** operate in continuous spaces $(x_t \in \mathbb{R}^n)$ where forward and reverse processes are characterized by Gaussian distributions:

$$q\left(x_t|x_{t-1}\right) := \mathcal{N}\left(x_t; \sqrt{1-\beta_t}x_{t-1}, \beta_t I\right)$$
$$q\left(x_T\right) := \mathcal{N}\left(x_T; 0, I\right)$$
$$p_\theta\left(x_{t-1}|x_t\right) := \mathcal{N}\left(x_{t-1}; \mu_\theta\left(x_t, t\right), \Sigma_\theta\left(x_t, t\right)\right)$$

Ho et al. (2020) suggest using diagonal $\Sigma_\theta\left(x_t, t\right)$ with a constant $\sigma_t$ and computing $\mu_\theta\left(x_t, t\right)$ as a function of $x_t$ and $\epsilon_\theta(x_t, t)$:

$$\mu_\theta\left(x_t, t\right) = \frac{1}{\sqrt{\alpha_t}}\left(x_t - \frac{\beta_t}{\sqrt{1-\bar{\alpha}_t}}\epsilon_\theta\left(x_t, t\right)\right)$$

where $\alpha_t := 1 - \beta_t$, $\bar{\alpha}_t := \prod_{i\leq t}\alpha_i$ and $\epsilon_\theta(x_t, t)$ predicts a "groundtruth" noise component $\epsilon$ for the noisy data sample $x_t$. In practice, the objective (1) can be simplified to the sum of mean-squared errors between $\epsilon_\theta(x_t, t)$ and $\epsilon$ over all timesteps $t$:

$$L_t^{simple} = \mathbb{E}_{x_0,\epsilon,t}\|\epsilon - \epsilon_\theta(x_t, t)\|_2^2$$
(2)

**Multinomial diffusion models** (Hoogeboom et al., 2021) are designed to generate categorical data where $x_t \in \{0, 1\}^K$ is a one-hot encoded categorical variable with $K$ values. The multinomial forward diffusion process defines $q\left(x_t|x_{t-1}\right)$ as a categorical distribution that corrupts the data by uniform noise over $K$ classes:

$$q(x_t|x_{t-1}) := Cat\left(x_t; (1-\beta_t)x_{t-1} + \beta_t/K\right)$$
$$q\left(x_T\right) := Cat\left(x_T; 1/K\right)$$
$$q\left(x_t|x_0\right) = Cat\left(x_t; \bar{\alpha}_t x_0 + (1-\bar{\alpha}_t)/K\right)$$

From the equations above, the posterior $q(x_{t-1}|x_t, x_0)$ can be derived:

$$q\left(x_{t-1}|x_t, x_0\right) = Cat\left(x_{t-1}; \pi/\sum_{k=1}^{K}\pi_k\right)$$

where $\pi = [\alpha_t x_t + (1-\alpha_t)/K] \odot [\bar{\alpha}_{t-1}x_0 + (1-\bar{\alpha}_{t-1})/K]$.

The reverse distribution $p_\theta\left(x_{t-1}|x_t\right)$ is parameterized as $q\left(x_{t-1}|x_t, \hat{x}_0(x_t, t)\right)$, where $\hat{x}_0$ is predicted by a neural network. Then, the model is trained to maximize the variational lower bound (1).

## 4 TABDDPM

In this section, we describe the design of TabDDPM as well as its main hyperparameters, which affect the model's effectiveness.

**TabDDPM** uses the multinomial diffusion to model the categorical and binary features, and the Gaussian diffusion to model the numerical ones. In more detail, for a tabular data sample $x = [x_{num}, x_{cat_1}, ..., x_{cat_C}]$, that consists of $N_{num}$ numerical features $x_{num} \in \mathbb{R}^{N_{num}}$ and $C$ categorical features $x_{cat_i}$ with $K_i$ categories each, our model takes one-hot encoded versions of categorical features as an input (i.e. $x_{cat_i}^{ohe} \in \{0, 1\}^{K_i}$) and normalized numerical features. Therefore, the input $x_0$ has a dimensionality of $(N_{num} + \sum K_i)$. For preprocessing, we use the gaussian

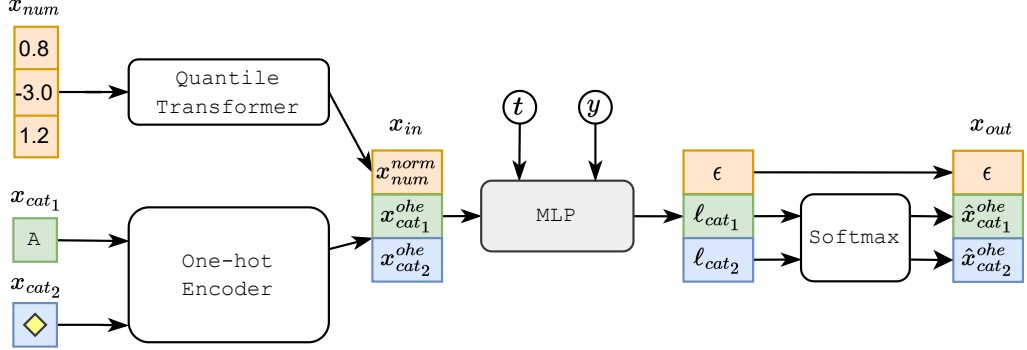

Figure 1: TabDDPM scheme for classification problems; $t$, $y$ and $\ell$ denote a diffusion timestep, a class label, and logits, respectively.

quantile transformation from the scikit-learn library (Pedregosa et al., 2011). Each categorical feature is handled by a separate forward diffusion process, i.e., the noise components for all features are sampled independently. The reverse diffusion step in TabDDPM is modelled by a multi-layer neural network that has an output of the same dimensionality as $x_0$, where the first $N_{num}$ coordinates are the predictions of $\epsilon$ for the Gaussian diffusion and the rest are the predictions of $x_{cat_i}^{ohe}$ for the multinomial diffusions.

The TabDDPM model for the classification problems is schematically presented on Figure 1. The model is trained by minimizing a sum of mean-squared error $L_t^{simple}$ (Equation (2)) for the Gaussian diffusion term and the KL divergences $L_t^i$ for each multinomial diffusion term (Equation (1)). The total loss of multinomial diffusions is additionally divided by the number of categorical features.

$$L_t^{TabDDPM} = L_t^{simple} + \frac{\sum_{i \leq C} L_t^i}{C} \quad (3)$$

For classification datasets, we use a class-conditional model, i.e. $p_\theta(x_{t-1}|x_t, y)$ is learned. For regression datasets, we consider a target value as an additional numerical feature, and the joint distribution is learned.

| Hyperparameter | Search space |
|---|---|
| Learning rate | $\mathrm{LogUniform}[0.00001, 0.003]$ |
| Batch size | $\mathrm{Cat}\{256, 4096\}$ |
| Diffusion timesteps | $\mathrm{Cat}\{100, 1000\}$ |
| Training iterations | $\mathrm{Cat}\{5000, 10000, 20000\}$ |
| # MLP layers | $\mathrm{Int}\{2, 4, 6, 8\}$ |
| MLP width of layers | $\mathrm{Int}\{128, 256, 512, 1024\}$ |
| Proportion of samples | $\mathrm{Float}\{0.25, 0.5, 1, 2, 4, 8\}$ |
| Dropout | 0.0 |
| Scheduler | cosine (Nichol, 2021) |
| Gaussian diffusion loss | MSE |
| Number of tuning trials | 50 |

Table 1: The main hyperparameters of TabDDPM.

To model the reverse process, we use a simple MLP architecture adapted from (Gorishniy et al., 2021):

$$\mathrm{MLP}(x) = \mathrm{Linear}\left(\mathrm{MLPBlock}\left(\ldots\left(\mathrm{MLPBlock}(x)\right)\right)\right)$$
$$\mathrm{MLPBlock}(x) = \mathrm{Dropout}(\mathrm{ReLU}(\mathrm{Linear}(x))) \quad (4)$$

As in (Nichol, 2021; Dhariwal & Nichol, 2021), a tabular input $x_{in}$, a timestep $t$, and a class label $y$ are processed as follows.

$$t\_emb = \mathrm{Linear}(\mathrm{SiLU}(\mathrm{Linear}(\mathrm{SinTimeEmb}(t))))$$
$$y\_emb = \mathrm{Embedding}(y) \quad (5)$$
$$x = \mathrm{Linear}(x_{in}) + t\_emb + y\_emb$$

where $\mathrm{SinTimeEmb}$ refers to a sinusoidal time embedding as in (Nichol, 2021; Dhariwal & Nichol, 2021) with a dimension of 128. All $\mathrm{Linear}$ layers in Equation 5 have a fixed projection dimension 128.

| Abbr | Name | # Train | # Validation | # Test | # Num | # Cat | Task type |
|------|------|---------|--------------|--------|-------|-------|-----------|
| AB | Abalone | 2672 | 669 | 836 | 7 | 1 | Regression |
| AD | Adult ROC | 26048 | 6513 | 16281 | 6 | 8 | Binclass |
| BU | Buddy | 12053 | 3014 | 3767 | 4 | 5 | Multiclass |
| CA | California Housing | 13209 | 3303 | 4128 | 8 | 0 | Regression |
| CAR | Cardio | 44800 | 11200 | 14000 | 5 | 6 | Binclass |
| CH | Churn Modelling | 6400 | 1600 | 2000 | 7 | 4 | Binclass |
| DE | Default | 19200 | 4800 | 6000 | 20 | 3 | Binclass |
| DI | Diabetes | 491 | 123 | 154 | 8 | 0 | Binclass |
| FB | Facebook Comments Volume | 157638 | 19722 | 19720 | 50 | 1 | Regression |
| GE | Gesture Phase | 6318 | 1580 | 1975 | 32 | 0 | Multiclass |
| HI | Higgs Small | 62751 | 15688 | 19610 | 28 | 0 | Binclass |
| HO | House 16H | 14581 | 3646 | 4557 | 16 | 0 | Regression |
| IN | Insurance | 856 | 214 | 268 | 3 | 3 | Regression |
| KI | King | 13832 | 3458 | 4323 | 17 | 3 | Regression |
| MI | MiniBooNE | 83240 | 20811 | 26013 | 50 | 0 | Binclass |
| WI | Wilt | 3096 | 775 | 968 | 5 | 0 | Binclass |

Table 2: List of datasets used for the evaluation and their descriptions.

**Hyperparameters** in TabDDPM are essential since in the experiments we observed them having a strong influence on the model effectiveness. Table 1 lists the main hyperparameters as well as the search spaces for each of them, which we recommend to use. The process of tuning is described in detail in the experimental section.

## 5 EXPERIMENTS

In this section, we extensively evaluate TabDDPM against existing alternatives.

**Datasets.** For systematic investigation of the performance of tabular generative models, we consider a diverse set of 15 real-world public datasets. These datasets have various sizes, nature, number of features, and their distributions. Most datasets were previously used for tabular model evaluation in (Zhao et al., 2021; Gorishniy et al., 2021). The full list of datasets and their properties are presented in Table 2.

**Baselines.** Since the number of generative models proposed for tabular data is enormous, we evaluate TabDDPM only against the leading approaches from each paradigm of generative modelling. Also, we consider only the baselines with the published source code.

- **TVAE** (Xu et al., 2019) — the state-of-the-art variational auto-encoder for tabular data generation. To the best of our knowledge, there are no alternative VAE-like models that outperform TVAE and have public source code.

- **CTABGAN** (Zhao et al., 2021) — a recent GAN-based model that is shown to outperform the existing tabular GANs on a diverse set of benchmarks. This approach cannot handle regression tasks.

- **CTABGAN+**(Zhao et al., 2022) — an extension of the **CTABGAN** model that was published in the very recent preprint. We are not aware of the GAN-based model for tabular data that is proposed after **CTABGAN+** and has a public source code.

- **SMOTE** (Chawla et al., 2002) — a "shallow" interpolation-based method that "generates" a synthetic point as a convex combination of a real data point and its $k$-th nearest neighbor from the dataset. This method was originally proposed for minor class oversampling. Here, we generalize it and apply it to synthetic data generation as a simple sanity check.

**Evaluation measure.** Our primary evaluation measure is *machine learning (ML) efficiency* (or utility) (Xu et al., 2019). In more detail, ML efficiency quantifies the performance of classification or regression models that are trained on synthetic data and evaluated on the real test set. Intuitively, models trained on high-quality synthetics should be competitive (or even superior) to models trained

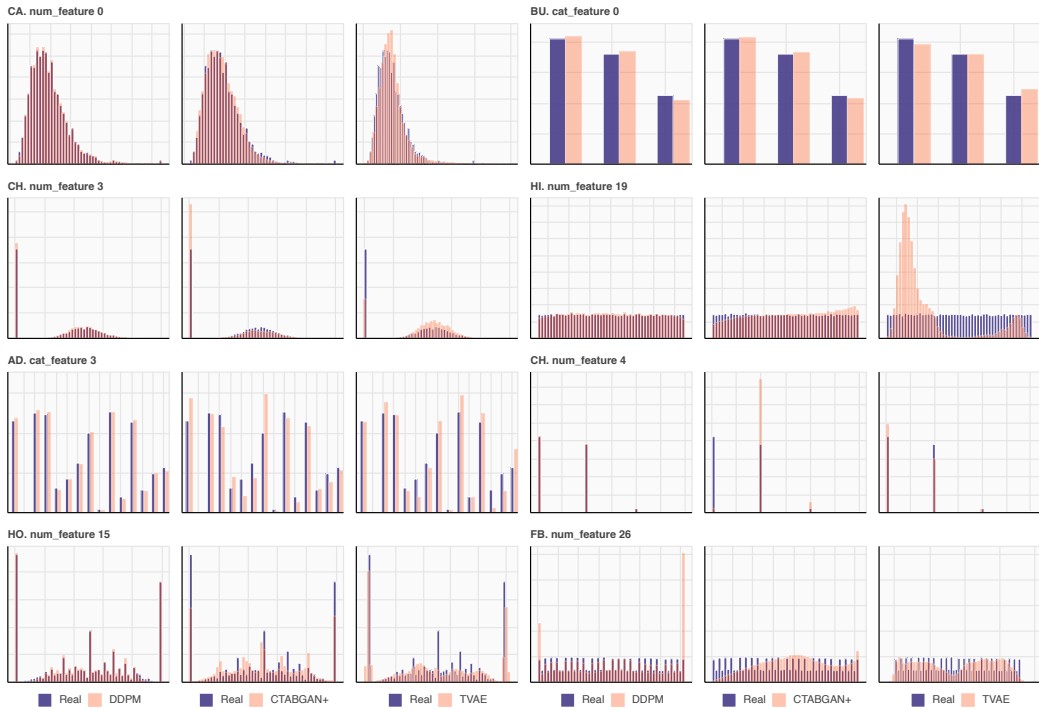

Figure 2: The individual feature distributions for the real data and the data generated by TabDDPM, CTABGAN+, and TVAE. TabDDPM produces more realistic feature distributions than alternatives in most cases.

on real data. In our experiments, we use two evaluation protocols to compute ML efficiency. In the first protocol, which is more common in the literature (Xu et al., 2019; Zhao et al., 2021; Kim et al., 2022), we compute an average efficiency with respect to a set of diverse ML models (logistic regression, decision tree, and others). In the second protocol, we evaluate ML efficiency only with respect to the CatBoost model (Prokhorenkova et al., 2018), which is arguably the leading GBDT implementation providing state-of-the-art performance on tabular tasks Gorishniy et al. (2021). In our experiments in subsection 5.2, we show that it is crucial to use the second protocol, while the first one can often be misleading.

**Tuning process.** To tune the hyperparameters of TabDDPM and the baselines, we use the Optuna library (Akiba et al., 2019). The tuning process is guided by the values of the ML efficiency (with respect to Catboost) of the generated synthetic data on a hold-out validation dataset (the score is averaged over five different sampling seeds). The search spaces for all hyperparameters of TabDDPM are reported in Table 1 (for baselines — in Appendix C). Additionally, we demonstrate that tuning the hyperparameters using the CatBoost guidance does not introduce any sort of "Catboost-biasedness", and the Catboost-tuned TabDDPM produces synthetics that are also superior for other models, like MLP. These results are reported in Appendix A.

### 5.1 QUALITATIVE COMPARISON

Here, we qualitatively investigate the ability of TabDDPM to model the individual and joint feature distributions compared to the TVAE and CTABGAN+ baselines. In particular, for each dataset, we sample a synthetic dataset from TabDDPM, TVAE, and CTABGAN+ of the same size as a real train set in a particular dataset. For classification datasets, each class is sampled according to its proportion in the real dataset. Then, we visualize the typical individual feature distributions for real and synthetic data in Figure 2. For completeness, the features of different types and distributions are presented. In most cases, TabDDPM produces more realistic feature distributions compared to TVAE and CTABGAN+. The advantage is more pronounced (1) for numerical features, which are uniformly distributed, (2) for categorical features with high cardinality, and (3) for mixed type features

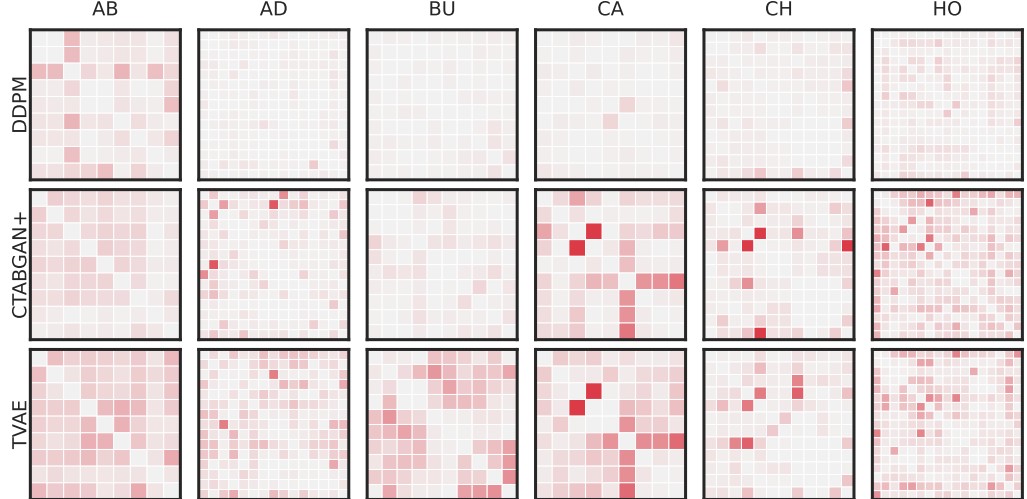

Figure 3: The absolute difference between correlation matrices computed on real and synthetic datasets. A more intensive red colour indicates a higher difference between the real and synthetic correlation values.

that combine continuous and discrete distributions. Then, we also visualize the differences between the correlation matrices computed on real and synthetic data for different datasets, see Figure 3. To compute the correlation matrices, we use the Pearson correlation coefficient for numerical-numerical correlations, the correlation Ratio for categorical-numerical cases, and Theil's U statistic between categorical features. Compared to CTABGAN+ and TVAE, TabDDPM generates synthetic datasets with more realistic pairwise correlations. These illustrations indicate that our TabDDPM model is more flexible compared to alternatives and produces superior synthetic data.

## 5.2 MACHINE LEARNING EFFICIENCY

In this section, we compare TabDDPM to alternative generative models in terms of machine learning efficiency. From each generative model, we sample a synthetic dataset with the size of a real train set in proportion from Table 1. This synthetic data is then used to train a classification/regression model, which is then evaluated using the real test set. In our experiments, classification performance is evaluated by the F1 score, and regression performance is evaluated by the R2 score. We use two protocols:

1. First, we compute average ML efficiency for a diverse set of ML models, as performed in previous works (Xu et al., 2019; Zhao et al., 2021; Kim et al., 2022). This set includes Decision Tree, Random Forest, Logistic Regression (or Ridge Regression) and MLP models from the scikit-learn library (Pedregosa et al., 2011) with the default hyperparameters except for "max-depth" equals 28 for Decision Tree and Random Forest, "maximum iterations" equals 500 for Logistic and Ridge regressions, and "maximum iterations" equals 100 for MLPs.

2. Second, we compute ML efficiency with respect to the current state-of-the-art model for tabular data. Specifically, we consider CatBoost (Prokhorenkova et al., 2018) and MLP architecture from (Gorishniy et al., 2021) for evaluation. CatBoost and MLP hyperparameters are thoroughly tuned on each dataset using the search spaces from (Gorishniy et al., 2021). We argue that this evaluation protocol demonstrates the practical value of synthetic data more reliably since in most real scenarios practitioners are not interested in using weak and suboptimal classifiers/regressors.

**Main results.** The ML efficiency values computed by both protocols are presented in Table 3 and in Table 4. The ML efficiency for the tuned MLP is reported in Appendix A. To compute each value, we average the results over five random seeds for synthetics generation, and for each generated

|  | AB ($R2$) | AD ($F1$) | BU ($F1$) | CA ($R2$) | CAR ($F1$) | CH ($F1$) | DE ($F1$) | DI ($F1$) |
|---|---|---|---|---|---|---|---|---|
| TVAE | $0.238_{\pm.012}$ | $0.742_{\pm.001}$ | $0.779_{\pm.004}$ | $-13.0_{\pm1.51}$ | $0.693_{\pm.002}$ | $0.684_{\pm.003}$ | $0.643_{\pm.003}$ | $0.712_{\pm.010}$ |
| CTABGAN | – | $0.737_{\pm.007}$ | $0.786_{\pm.008}$ | – | $0.684_{\pm.003}$ | $0.636_{\pm.010}$ | $0.614_{\pm.007}$ | $0.655_{\pm.015}$ |
| CTABGAN+ | $0.316_{\pm.024}$ | $0.730_{\pm.007}$ | $0.837_{\pm.006}$ | $-7.59_{\pm.645}$ | $\mathbf{0.708}_{\pm.002}$ | $0.650_{\pm.008}$ | $0.648_{\pm.008}$ | $\mathbf{0.727}_{\pm.023}$ |
| SMOTE | $\mathbf{0.400}_{\pm.009}$ | $0.750_{\pm.004}$ | $0.842_{\pm.003}$ | $0.667_{\pm.006}$ | $0.693_{\pm.001}$ | $0.690_{\pm.004}$ | $0.649_{\pm.003}$ | $0.677_{\pm.013}$ |
| TabDDPM | $\mathbf{0.392}_{\pm.009}$ | $\mathbf{0.758}_{\pm.005}$ | $\mathbf{0.851}_{\pm.003}$ | $\mathbf{0.695}_{\pm.002}$ | $0.696_{\pm.001}$ | $\mathbf{0.693}_{\pm.003}$ | $\mathbf{0.659}_{\pm.003}$ | $0.675_{\pm.011}$ |
| Real | $0.423_{\pm.009}$ | $0.750_{\pm.006}$ | $0.845_{\pm.004}$ | $0.663_{\pm.002}$ | $0.683_{\pm.002}$ | $0.679_{\pm.003}$ | $0.648_{\pm.003}$ | $0.699_{\pm.012}$ |

|  | FB ($R2$) | GE ($F1$) | HI ($F1$) | HO ($R2$) | IN ($R2$) | KI ($R2$) | MI ($F1$) | WI ($F1$) |
|---|---|---|---|---|---|---|---|---|
| TVAE | $\ll 0$ | $0.372_{\pm.006}$ | $0.590_{\pm.004}$ | $0.174_{\pm.012}$ | $0.470_{\pm.024}$ | $0.666_{\pm.006}$ | $\mathbf{0.880}_{\pm.002}$ | $0.497_{\pm.001}$ |
| CTABGAN | – | $0.339_{\pm.009}$ | $0.539_{\pm.006}$ | – | – | – | $0.856_{\pm.003}$ | $0.656_{\pm.011}$ |
| CTABGAN+ | $\ll 0$ | $0.373_{\pm.009}$ | $0.598_{\pm.004}$ | $0.222_{\pm.042}$ | $0.669_{\pm.018}$ | $0.197_{\pm.051}$ | $0.867_{\pm.002}$ | $0.653_{\pm.027}$ |
| SMOTE | $\mathbf{0.651}_{\pm.002}$ | $\mathbf{0.478}_{\pm.005}$ | $0.664_{\pm.003}$ | $0.394_{\pm.006}$ | $0.709_{\pm.008}$ | $\mathbf{0.751}_{\pm.005}$ | $0.860_{\pm.001}$ | $\mathbf{0.793}_{\pm.004}$ |
| TabDDPM | $0.527_{\pm.005}$ | $0.462_{\pm.005}$ | $\mathbf{0.670}_{\pm.002}$ | $\mathbf{0.426}_{\pm.007}$ | $\mathbf{0.734}_{\pm.007}$ | $0.611_{\pm.013}$ | $0.850_{\pm.004}$ | $\mathbf{0.792}_{\pm.004}$ |
| Real | $0.645_{\pm.005}$ | $0.431_{\pm.005}$ | $0.663_{\pm.002}$ | $0.415_{\pm.007}$ | $0.708_{\pm.007}$ | $0.768_{\pm.013}$ | $0.850_{\pm.004}$ | $0.684_{\pm.004}$ |

Table 3: The values of machine learning efficiency computed with regards to five weak classification/regression models. Negative scores denote negative R2, which means that performance is worse than an optimal constant prediction.

|  | AB ($R2$) | AD ($F1$) | BU ($F1$) | CA ($R2$) | CAR ($F1$) | CH ($F1$) | DE ($F1$) | DI ($F1$) |
|---|---|---|---|---|---|---|---|---|
| TVAE | $0.433_{\pm.008}$ | $0.781_{\pm.002}$ | $0.864_{\pm.005}$ | $0.752_{\pm.001}$ | $0.717_{\pm.001}$ | $0.732_{\pm.006}$ | $0.656_{\pm.007}$ | $\mathbf{0.714}_{\pm.039}$ |
| CTABGAN | – | $0.783_{\pm.002}$ | $0.855_{\pm.005}$ | – | $0.717_{\pm.001}$ | $0.688_{\pm.006}$ | $0.644_{\pm.011}$ | $\mathbf{0.731}_{\pm.022}$ |
| CTABGAN+ | $0.467_{\pm.004}$ | $0.772_{\pm.003}$ | $0.884_{\pm.005}$ | $0.525_{\pm.004}$ | $0.733_{\pm.001}$ | $0.702_{\pm.012}$ | $0.686_{\pm.004}$ | $\mathbf{0.734}_{\pm.020}$ |
| SMOTE | $0.549_{\pm.005}$ | $0.791_{\pm.002}$ | $0.891_{\pm.004}$ | $\mathbf{0.840}_{\pm.001}$ | $0.732_{\pm.001}$ | $0.743_{\pm.005}$ | $\mathbf{0.693}_{\pm.003}$ | $0.683_{\pm.037}$ |
| TabDDPM | $\mathbf{0.550}_{\pm.010}$ | $\mathbf{0.795}_{\pm.001}$ | $\mathbf{0.906}_{\pm.003}$ | $0.836_{\pm.002}$ | $\mathbf{0.737}_{\pm.001}$ | $\mathbf{0.755}_{\pm.006}$ | $0.691_{\pm.004}$ | $\mathbf{0.740}_{\pm.020}$ |
| Real | $0.556_{\pm.004}$ | $0.815_{\pm.002}$ | $0.906_{\pm.002}$ | $0.857_{\pm.001}$ | $0.738_{\pm.001}$ | $0.740_{\pm.009}$ | $0.688_{\pm.003}$ | $0.785_{\pm.013}$ |

|  | FB ($R2$) | GE ($F1$) | HI ($F1$) | HO ($R2$) | IN ($R2$) | KI ($R2$) | MI ($F1$) | WI ($F1$) |
|---|---|---|---|---|---|---|---|---|
| TVAE | $0.685_{\pm.003}$ | $0.434_{\pm.006}$ | $0.638_{\pm.003}$ | $0.493_{\pm.006}$ | $0.784_{\pm.010}$ | $0.824_{\pm.003}$ | $0.912_{\pm.001}$ | $0.501_{\pm.012}$ |
| CTABGAN | – | $0.392_{\pm.006}$ | $0.575_{\pm.006}$ | – | – | – | $0.889_{\pm.002}$ | $\mathbf{0.906}_{\pm.019}$ |
| CTABGAN+ | $0.509_{\pm.011}$ | $0.406_{\pm.009}$ | $0.664_{\pm.002}$ | $0.504_{\pm.005}$ | $0.797_{\pm.005}$ | $0.444_{\pm.014}$ | $0.892_{\pm.002}$ | $0.798_{\pm.021}$ |
| SMOTE | $\mathbf{0.803}_{\pm.002}$ | $\mathbf{0.658}_{\pm.007}$ | $\mathbf{0.722}_{\pm.001}$ | $0.662_{\pm.004}$ | $\mathbf{0.812}_{\pm.002}$ | $\mathbf{0.842}_{\pm.004}$ | $0.932_{\pm.001}$ | $\mathbf{0.913}_{\pm.007}$ |
| TabDDPM | $0.713_{\pm.002}$ | $0.597_{\pm.006}$ | $\mathbf{0.722}_{\pm.001}$ | $\mathbf{0.677}_{\pm.010}$ | $0.809_{\pm.002}$ | $\mathbf{0.833}_{\pm.014}$ | $\mathbf{0.936}_{\pm.001}$ | $0.904_{\pm.009}$ |
| Real | $0.837_{\pm.001}$ | $0.636_{\pm.007}$ | $0.724_{\pm.001}$ | $0.662_{\pm.003}$ | $0.814_{\pm.001}$ | $0.907_{\pm.002}$ | $0.934_{\pm.000}$ | $0.898_{\pm.006}$ |

Table 4: The values of machine learning efficiency computed with regards to the state-of-the-art tuned CatBoost model.

dataset, we average over ten random seeds for training classifiers/regressors. The key observations are described below:

- In both evaluation protocols, TabDDPM significantly outperforms TVAE and CTABGAN+ on most datasets, which highlights the advantage of diffusion models for tabular data as well as demonstrated for other domains in prior works.

- The interpolation-based SMOTE method demonstrates the performance competitive to TabDDPM and often significantly outperforms the GAN/VAE approaches. Interestingly, most of the prior works on generative models for tabular data do not compare against SMOTE, while it appears to be a simple baseline, which is challenging to beat.

- While many prior works use the first evaluation protocol to compute the ML efficiency, we argue that the second one (which uses the state-of-the-art model, like CatBoost) is more appropriate. Table 3 and Table 4 show that the absolute values of classification/regression performance are much lower for the first protocol, i.e., weak classifiers/regressors are substantially inferior to CatBoost on the considered benchmarks. Therefore, one can hardly use these suboptimal models instead of CatBoost and their performance values are uninformative for practitioners. Moreover, in the first protocol, training on synthetic data is often advantageous compared to training on real data. This creates an impression that the data produced by generative models are more valuable than the real ones. However, it is not the case when one uses the tuned ML model, as in most practical scenarios. Appendix A confirms this observation for the properly tuned MLP model.

Overall, TabDDPM provides state-of-the-art generative performance and can be used as a source of high-quality synthetic data. Interestingly, in terms of ML efficiency, a simple "shallow" SMOTE method is competitive to TabDDPM, which raises the question if sophisticated deep generative models are needed. In the section below, we provide an affirmative answer to this question.

## 5.3 PRIVACY

Here, we demonstrate that TabDDPM is preferable to SMOTE in setups with privacy concerns, e.g., sharing the data without disclosure of personal or sensitive information. In these setups, one is interested in high-quality synthetics that do not reveal the datapoints from the original real dataset. To quantify the privacy of synthetic, we use a median Distance to Closest Record (DCR) (Zhao et al., 2021) between synthetic and real datapoints. Specifically, for each synthetic sample, we find the minimum distance to real datapoints and take the median of these distances. Low DCR values indicate that all synthetic samples are essentially copies of some real datapoints, which violates the privacy requirements. In contrast, larger DCR values indicate that the generative model can produce something "new" rather than just copies of real data. Table 5 compares the DCR values for SMOTE and TabDDPM and demonstrates the advantage of TabDDPM consistently for all datasets. We also visualize histograms of the minimal synthetic-to-real distances on Figure 4. For SMOTE, most distance values are concentrated around zero, while TabDDPM samples are better separated from real datapoints. This experiment confirms that TabDDPM synthetics while providing high ML efficiency, are also more appropriate for privacy-concerned scenarios.

|  | AB | | AD | | BU | | CA | | CAR | | CH | | DE | | DI | |
|---|---|---|---|---|---|---|---|---|---|---|---|---|---|---|---|---|
|  | score | DCR | score | DCR | score | DCR | score | DCR | score | DCR | score | DCR | score | DCR | score | DCR |
| SMOTE | 0.549 | 0.014 | 0.791 | 0.024 | 0.891 | 0.054 | 0.840 | 0.014 | 0.732 | 0.007 | 0.743 | 0.077 | 0.693 | 0.027 | 0.683 | 0.068 |
| TabDDPM | 0.550 | **0.050** | 0.795 | **0.104** | 0.906 | **0.143** | 0.836 | **0.041** | 0.737 | **0.012** | 0.755 | **0.157** | 0.691 | **0.112** | 0.740 | **0.204** |

|  | FB | | GE | | HI | | HO | | IN | | KI | | MI | | WI | |
|---|---|---|---|---|---|---|---|---|---|---|---|---|---|---|---|---|
|  | score | DCR | score | DCR | score | DCR | score | DCR | score | DCR | score | DCR | score | DCR | score | DCR |
| SMOTE | 0.803 | 0.027 | 0.658 | 0.023 | 0.722 | 0.319 | 0.662 | 0.056 | 0.812 | 0.030 | 0.842 | 0.066 | 0.932 | 0.016 | 0.913 | 0.007 |
| TabDDPM | 0.713 | **0.112** | 0.597 | **0.059** | 0.722 | **0.449** | 0.677 | **0.086** | 0.809 | **0.041** | 0.833 | **0.189** | 0.936 | **0.022** | 0.904 | **0.016** |

Table 5: ML efficiency CatBoost scores and privacy scores for SMOTE and TabDDPM models.

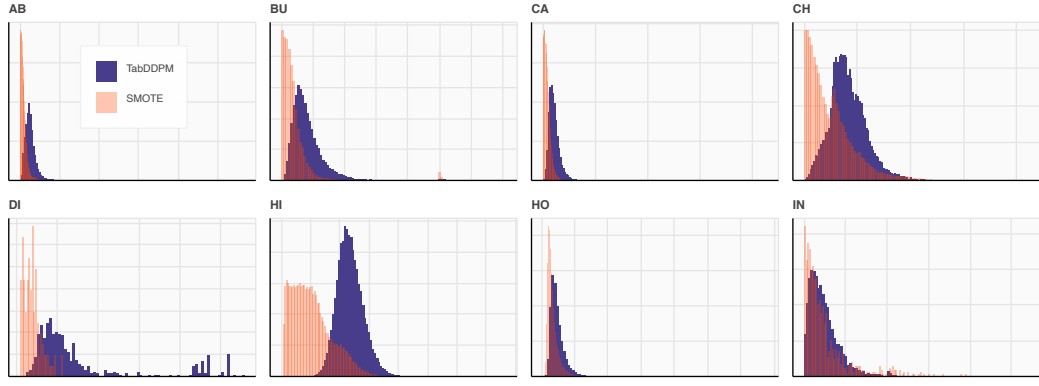

Figure 4: Histograms of minimal synthetic-to-real distances for TabDDM and SMOTE.

## 6 CONCLUSION

In this paper, we have investigated the prospect of the diffusion modelling framework in the field of tabular data. In particular, we have described the design of DDPM that can handle mixed data consisting of numerical, ordinal, and categorical features. We also demonstrate the importance of the model's hyperparameters and explain the protocol of their tuning. For the most considered benchmarks, the synthetics produced by our model has consistently higher quality compared to

ones produced by the GAN/VAE-based rivals and interpolation techniques, especially for the setups, where the privacy of the data must be ensured.

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

# APPENDIX

## A MLP EVALUATION AND TUNING

Here, we show that tuning the hyperparameters using the CatBoost guidance results in the TabDDPM models that produce synthetics that is also optimal for other classifiers/regressors. The results for a subset of datasets are presented on Table 6. The methods denoted with "-CB" and "-MLP" denote the CatBoost guidance and different types of evaluation (CatBoost and MLP, respectively). The "-MLP-tune" suffix stands for the MLP guidance tuning and MLP evaluation.

| | AB ($R2$) | AD ($F1$) | BU ($F1$) | CA ($R2$) | CAR ($F1$) | CH ($F1$) | DE ($F1$) | DI ($F1$) |
|---|---|---|---|---|---|---|---|---|
| TabDDPM-CB | $0.550\pm.010$ | $0.795\pm.001$ | $0.906\pm.003$ | $0.836\pm.002$ | $0.737\pm.001$ | $0.755\pm.006$ | $0.691\pm.004$ | $0.740\pm.020$ |
| Real-CB | $0.556\pm.004$ | $0.815\pm.002$ | $0.906\pm.002$ | $0.857\pm.001$ | $0.738\pm.001$ | $0.740\pm.009$ | $0.688\pm.003$ | $0.785\pm.013$ |
| TabDDPM-MLP | $0.569\pm.010$ | $0.794\pm.002$ | $0.903\pm.003$ | $0.809\pm.003$ | $0.737\pm.001$ | $0.750\pm.005$ | $0.679\pm.008$ | $0.754\pm.020$ |
| Real-MLP | $0.581\pm.005$ | $0.795\pm.001$ | $0.905\pm.003$ | $0.808\pm.002$ | $0.739\pm.001$ | $0.741\pm.006$ | $0.688\pm.004$ | $0.754\pm.017$ |
| TabDDPM-MLP-tune | $0.559\pm.009$ | $0.792\pm.002$ | $0.901\pm.003$ | $0.803\pm.004$ | $0.737\pm.001$ | $0.749\pm.006$ | $0.674\pm.013$ | $0.741\pm.018$ |

| | FB ($R2$) | GE ($F1$) | HI ($F1$) | HO ($R2$) | IN ($R2$) | KI ($R2$) | MI ($F1$) | WI ($F1$) |
|---|---|---|---|---|---|---|---|---|
| TabDDPM-CB | $0.713\pm.002$ | $0.597\pm.006$ | $0.722\pm.001$ | $0.677\pm.010$ | $0.809\pm.002$ | $0.833\pm.014$ | $0.936\pm.001$ | $0.904\pm.009$ |
| Real-CB | $0.837\pm.001$ | $0.636\pm.007$ | $0.724\pm.001$ | $0.662\pm.003$ | $0.814\pm.001$ | $0.907\pm.002$ | $0.934\pm.000$ | $0.898\pm.006$ |
| TabDDPM-MLP | – | $0.595\pm.006$ | $0.717\pm.002$ | $0.643\pm.010$ | $0.794\pm.008$ | $0.804\pm.015$ | $0.938\pm.001$ | $0.921\pm.006$ |
| Real-MLP | – | $0.607\pm.007$ | $0.717\pm.002$ | $0.614\pm.006$ | $0.800\pm.003$ | $0.882\pm.004$ | $0.936\pm.001$ | $0.905\pm.006$ |
| TabDDPM-MLP-tune | – | – | – | $0.626\pm.009$ | $0.800\pm.004$ | $0.799\pm.018$ | – | $0.914\pm.006$ |

Table 6: ML utility score with MLP evaluation and MLP tuning compared with CatBoost evaluation and CatBoost tuning.

## B ADDITIONAL RESULTS

Here, we provide results for CTGAN Xu et al. (2019) model (Table 7). We also follow Zhao et al. (2021) and provide an additional quantitative comparison that shows how well individual feature distributions are modelled (Table 8, Table 9, Table 10). Finally, we include density and coverage metrics from Naeem et al. (2020) that are improved alternatives of precision and recall, respectively (Table 11, Table 12).

| | AB ($R2$) | AD ($F1$) | BU ($F1$) | CA ($R2$) | CAR ($F1$) | CH ($F1$) | DE ($F1$) | DI ($F1$) |
|---|---|---|---|---|---|---|---|---|
| CTGAN | $0.420\pm.004$ | $0.789\pm.001$ | $0.867\pm.003$ | $0.686\pm.003$ | $0.730\pm.001$ | $0.723\pm.006$ | $\mathbf{0.699\pm.002}$ | $0.459\pm.096$ |
| TVAE | $0.433\pm.008$ | $0.781\pm.002$ | $0.864\pm.005$ | $0.752\pm.001$ | $0.717\pm.001$ | $0.732\pm.006$ | $0.656\pm.007$ | $\mathbf{0.714\pm.039}$ |
| CTABGAN | – | $0.783\pm.002$ | $0.855\pm.005$ | – | $0.717\pm.001$ | $0.688\pm.006$ | $0.644\pm.011$ | $\mathbf{0.731\pm.022}$ |
| CTABGAN+ | $0.467\pm.004$ | $0.772\pm.003$ | $0.884\pm.005$ | $0.525\pm.004$ | $0.733\pm.001$ | $0.702\pm.012$ | $0.686\pm.004$ | $\mathbf{0.734\pm.020}$ |
| SMOTE | $\mathbf{0.549\pm.005}$ | $0.791\pm.002$ | $0.891\pm.003$ | $\mathbf{0.840\pm.001}$ | $0.732\pm.001$ | $0.743\pm.005$ | $0.693\pm.003$ | $0.683\pm.037$ |
| TabDDPM | $\mathbf{0.550\pm.010}$ | $\mathbf{0.795\pm.001}$ | $\mathbf{0.906\pm.003}$ | $0.836\pm.002$ | $\mathbf{0.737\pm.001}$ | $\mathbf{0.755\pm.006}$ | $0.691\pm.004$ | $\mathbf{0.740\pm.020}$ |
| Real | $0.556\pm.004$ | $0.815\pm.002$ | $0.906\pm.002$ | $0.857\pm.001$ | $0.738\pm.001$ | $0.740\pm.009$ | $0.688\pm.003$ | $0.785\pm.013$ |

| | FB ($R2$) | GE ($F1$) | HI ($F1$) | HO ($R2$) | IN ($R2$) | KI ($R2$) | MI ($F1$) | WI ($F1$) |
|---|---|---|---|---|---|---|---|---|
| CTGAN | $0.443\pm.005$ | $0.333\pm.013$ | $0.575\pm.006$ | $0.433\pm.005$ | $0.745\pm.009$ | $0.772\pm.005$ | $0.783\pm.005$ | $0.749\pm.015$ |
| TVAE | $0.685\pm.003$ | $0.434\pm.006$ | $0.638\pm.003$ | $0.493\pm.006$ | $0.784\pm.010$ | $0.824\pm.003$ | $0.912\pm.001$ | $0.501\pm.012$ |
| CTABGAN | – | $0.392\pm.006$ | $0.575\pm.004$ | – | – | – | $0.889\pm.002$ | $\mathbf{0.906\pm.019}$ |
| CTABGAN+ | $0.509\pm.011$ | $0.406\pm.009$ | $0.664\pm.002$ | $0.504\pm.005$ | $0.797\pm.005$ | $0.444\pm.014$ | $0.892\pm.002$ | $0.798\pm.021$ |
| SMOTE | $\mathbf{0.803\pm.002}$ | $\mathbf{0.658\pm.007}$ | $\mathbf{0.722\pm.001}$ | $0.662\pm.004$ | $\mathbf{0.812\pm.002}$ | $\mathbf{0.842\pm.004}$ | $0.932\pm.001$ | $\mathbf{0.913\pm.007}$ |
| TabDDPM | $0.713\pm.002$ | $0.597\pm.006$ | $\mathbf{0.722\pm.001}$ | $\mathbf{0.677\pm.010}$ | $0.809\pm.002$ | $\mathbf{0.833\pm.014}$ | $\mathbf{0.936\pm.001}$ | $0.904\pm.009$ |
| Real | $0.837\pm.001$ | $0.636\pm.007$ | $0.724\pm.001$ | $0.662\pm.003$ | $0.814\pm.001$ | $0.907\pm.002$ | $0.934\pm.000$ | $0.898\pm.006$ |

Table 7: ML utility score with CatBoost evaluation.

|          | AB    | AD    | BU    | CA    | CA    | CH    | DE    | DI    |
|----------|-------|-------|-------|-------|-------|-------|-------|-------|
| CTGAN    | 0.008 | 0.010 | 0.015 | 0.004 | 0.004 | 0.009 | 0.004 | 0.085 |
| TVAE     | 0.020 | 0.016 | 0.039 | 0.007 | 0.027 | 0.049 | 0.009 | 0.044 |
| CTABGAN+ | 0.008 | 0.011 | 0.016 | 0.019 | 0.003 | 0.046 | 0.022 | 0.016 |
| SMOTE    | **0.002** | 0.003 | 0.005 | **0.002** | 0.001 | 0.006 | **0.002** | 0.020 |
| TabDDPM  | 0.005 | **0.002** | **0.003** | 0.002 | **0.000** | 0.005 | 0.012 | **0.008** |

|          | FB    | GE    | HI    | HO    | IN    | KI    | MI    | WI    |
|----------|-------|-------|-------|-------|-------|-------|-------|-------|
| CTGAN    | 0.004 | 0.010 | **0.003** | 0.005 | 0.021 | 0.022 | 0.004 | 0.013 |
| TVAE     | 0.008 | 0.009 | 0.076 | 0.007 | 0.025 | 0.012 | 0.004 | 0.016 |
| CTABGAN+ | 0.078 | 0.007 | 0.052 | 0.008 | 0.025 | 0.021 | 0.006 | 0.006 |
| SMOTE    | **0.000** | **0.004** | 0.009 | 0.005 | 0.011 | **0.004** | **0.000** | **0.002** |
| TabDDPM  | 0.089 | 0.011 | 0.003 | **0.004** | **0.006** | 0.014 | 0.001 | **0.002** |

Table 8: Wasserstein distance between numerical features.

|          | AB    | AD    | BU    | CA    | CA    | CH    | DE    | DI    |
|----------|-------|-------|-------|-------|-------|-------|-------|-------|
| CTGAN    | 0.276 | 0.085 | 0.168 | *nan* | 0.076 | 0.039 | 0.120 | 0.298 |
| TVAE     | 0.027 | 0.095 | 0.072 | *nan* | 0.181 | 0.019 | 0.157 | 0.052 |
| CTABGAN+ | 0.035 | 0.052 | 0.037 | *nan* | **0.009** | 0.018 | 0.030 | 0.017 |
| SMOTE    | **0.005** | 0.074 | 0.072 | *nan* | 0.069 | 0.030 | 0.058 | **0.004** |
| TabDDPM  | 0.007 | **0.019** | **0.026** | *nan* | 0.011 | **0.017** | **0.009** | 0.006 |

|          | FB    | GE    | HI    | HO    | IN    | KI    | MI    | WI    |
|----------|-------|-------|-------|-------|-------|-------|-------|-------|
| CTGAN    | **0.017** | 0.240 | 0.091 | *nan* | 0.071 | 0.296 | 0.140 | 0.532 |
| TVAE     | 0.246 | 0.113 | 0.040 | *nan* | 0.033 | 0.098 | 0.066 | 0.149 |
| CTABGAN+ | 0.051 | 0.094 | 0.009 | *nan* | 0.023 | **0.044** | 0.075 | 0.017 |
| SMOTE    | 0.027 | **0.000** | **0.000** | *nan* | 0.013 | 0.102 | **0.000** | **0.000** |
| TabDDPM  | 0.046 | 0.001 | 0.001 | *nan* | **0.008** | 0.060 | **0.000** | 0.002 |

Table 9: Jensen-Shannon divergence between categorical features.

|          | AB    | AD    | BU    | CA    | CA    | CH    | DE    | DI    |
|----------|-------|-------|-------|-------|-------|-------|-------|-------|
| CTGAN    | 0.471 | 0.390 | 0.492 | 0.606 | 0.712 | 0.239 | 1.355 | 1.735 |
| TVAE     | 0.517 | 0.636 | 0.569 | 0.753 | 2.437 | 0.564 | 1.965 | 0.736 |
| CTABGAN+ | 0.283 | 0.576 | 0.164 | 0.749 | 0.738 | 0.727 | 1.496 | **0.435** |
| SMOTE    | **0.185** | 0.482 | 0.245 | 0.127 | 0.599 | **0.147** | **0.642** | 0.838 |
| TabDDPM  | 0.333 | **0.133** | **0.068** | **0.090** | **0.202** | 0.161 | 0.934 | 0.186 |

|          | FB     | GE    | HI    | HO    | IN    | KI    | MI     | WI    |
|----------|--------|-------|-------|-------|-------|-------|--------|-------|
| CTGAN    | 5.651  | 5.301 | 1.413 | 0.742 | 0.196 | 1.530 | 43.815 | 0.538 |
| TVAE     | 5.960  | 2.996 | 2.759 | 0.902 | 0.224 | 1.004 | 44.692 | 0.550 |
| CTABGAN+ | 6.782  | 1.977 | 1.241 | 0.978 | 0.207 | 3.898 | 31.704 | 0.319 |
| SMOTE    | **1.596** | **0.560** | 0.354 | 0.452 | 0.301 | **0.569** | **0.258** | **0.059** |
| TabDDPM  | 16.120 | 1.192 | **0.233** | **0.336** | **0.077** | 3.623 | 9.185 | 0.375 |

Table 10: L2 distance between correlation matrices.

|          | AB    | AD    | BU    | CA    | CA    | CH    | DE    | DI    |
|----------|-------|-------|-------|-------|-------|-------|-------|-------|
| CTGAN    | 0.224 | 0.708 | 0.780 | 0.586 | 0.938 | 0.865 | 0.698 | 0.238 |
| TVAE     | 0.347 | 1.126 | 1.032 | 0.746 | 0.845 | 1.043 | 0.808 | **1.565** |
| CTABGAN+ | 0.380 | 0.867 | 0.998 | 0.569 | 0.957 | 0.974 | 0.730 | 0.974 |
| SMOTE    | **1.389** | **1.415** | **1.226** | **1.329** | **1.200** | **1.238** | **1.282** | 1.413 |
| TabDDPM  | 0.904 | 1.008 | 1.116 | 1.027 | 1.011 | 1.148 | 0.810 | 0.831 |

|          | FB    | GE    | HI    | HO    | IN    | KI    | MI    | WI    |
|----------|-------|-------|-------|-------|-------|-------|-------|-------|
| CTGAN    | 0.147 | 0.035 | 0.702 | 0.467 | 0.927 | 0.719 | 0.361 | 0.763 |
| TVAE     | 0.005 | 0.248 | 0.960 | 0.604 | 1.072 | 0.868 | 0.747 | 0.919 |
| CTABGAN+ | 0.187 | 0.448 | 0.730 | 0.565 | 1.052 | 0.186 | 0.110 | 0.831 |
| SMOTE    | **0.926** | **1.531** | **1.682** | **1.595** | **1.213** | **1.335** | **1.308** | **1.251** |
| TabDDPM  | 0.633 | 1.460 | 1.152 | 1.195 | 1.150 | 0.884 | 0.972 | 1.009 |

Table 11: Density of synthetic data.

|        | AB    | AD    | BU    | CA    | CA    | CH    | DE    | DI    |
|--------|-------|-------|-------|-------|-------|-------|-------|-------|
| CTGAN    | 0.654 | 0.948 | 0.966 | 0.759 | 0.920 | 1.000 | 0.777 | 0.572 |
| TVAE     | 0.769 | 0.886 | 0.585 | 0.922 | 0.208 | 0.991 | 0.672 | 0.978 |
| CTABGAN+ | 0.960 | 0.951 | 0.999 | 0.459 | 0.960 | 0.830 | 0.841 | **1.000** |
| SMOTE    | **1.000** | 0.970 | 0.968 | **1.000** | 0.866 | 1.000 | 0.962 | 0.841 |
| TabDDPM  | **1.000** | **0.994** | **1.000** | 0.998 | **0.978** | 1.000 | **0.967** | 0.955 |

|        | FB    | GE    | HI    | HO    | IN    | KI    | MI    | WI    |
|--------|-------|-------|-------|-------|-------|-------|-------|-------|
| CTGAN    | 0.238 | 0.029 | 0.871 | 0.839 | 0.986 | 0.739 | 0.576 | 0.986 |
| TVAE     | 0.014 | 0.669 | 0.255 | 0.875 | 0.987 | 0.874 | 0.823 | 0.867 |
| CTABGAN+ | 0.222 | 0.640 | 0.557 | 0.952 | **1.000** | 0.479 | 0.241 | 0.994 |
| SMOTE    | **0.928** | **1.000** | **0.999** | **1.000** | 0.995 | 0.945 | **0.991** | **1.000** |
| TabDDPM  | 0.782 | 0.997 | 0.980 | **1.000** | **1.000** | **0.969** | 0.956 | **1.000** |

Table 12: Coverage of synthetic data.

## C HYPERPARAMETERS SEARCH SPACES

| Parameter | Distribution |
|-----------|--------------|
| Max depth | $\mathrm{UniformInt}[3, 10]$ |
| Learning rate | $\mathrm{LogUniform}[1e\text{-}5, 1]$ |
| Bagging temperature | $\mathrm{Uniform}[0, 1]$ |
| L2 leaf reg | $\mathrm{LogUniform}[1, 10]$ |
| Leaf estimation iterations | $\mathrm{UniformInt}[1, 10]$ |
| Number of tuning trials | 100 |

Table 13: CatBoost hyperparameters space from Gorishniy et al. (2021)

| Parameter | Distribution |
|-----------|--------------|
| # Layers | $\mathrm{UniformInt}[1, 8]$ |
| Layer size | $\mathrm{Int}\{64, 128, 256, 512, 1024\}$ |
| Dropout | $\{0, \mathrm{Uniform}[0, 0.5]\}$ |
| Learning rate | $\mathrm{LogUniform}[1e\text{-}5, 1e\text{-}2]$ |
| Weight decay | $\{0, \mathrm{LogUniform}[1e\text{-}6, 1e\text{-}3]\}$ |
| Number of tuning trials | 100 |

Table 14: MLP hyperparameters space from Gorishniy et al. (2021)

| Parameter | Distribution |
|-----------|--------------|
| k_neighbours | $\mathrm{Int}[5, 20]$ |
| $\lambda_{range}$ | $\mathrm{Float}[0, 1]$ |
| Proportion of samples | $\mathrm{Float}\{0.25, 0.5, 1, 2, 4, 8\}$ |
| Number of tuning trials | 50 |

Table 15: SMOTE hyperparameters search space. $\lambda_{range}$ denotes the range of interpolation coefficient to sample from

| Parameter | Distribution |
|---|---|
| # claassif. layers | $\text{UniformInt}[1, 4]$ |
| Classif. layer size | $\text{Int}\{64, 128, 256\}$ |
| Training iterations | $\text{Cat}\{1000, 5000, 10000\}$ |
| Batch Size | $\text{Int}\{512, 1024, 2048\}$ |
| random_dim | $\text{Int}\{16, 32, 64, 128\}$ |
| num_channels | $\text{Int}\{16, 32, 64\}$ |
| Proportion of samples | $\text{Float}\{0.25, 0.5, 1, 2, 4, 8\}$ |
| Number of tuning trials | 35 |

Table 16: CTABGAN and CTABGAN+ hyperparameters search space. See an official implementation[2]

| Parameter | Distribution |
|---|---|
| # claassif. layers | $\text{UniformInt}[1, 6]$ |
| Classif. layer size | $\text{Int}\{64, 128, 256, 512\}$ |
| Training iterations | $\text{Cat}\{5000, 20000, 30000\}$ |
| Batch Size | $\text{Cat}\{456, 4096\}$ |
| embedding_dim | $\text{Int}\{16, 32, 64, 128, 256, 512, 1024\}$ |
| loss factor | $\text{LogUniform}[0.01, 10]$ |
| Proportion of samples | $\text{Float}\{0.25, 0.5, 1, 2, 4, 8\}$ |
| Number of tuning trials | 50 |

Table 17: TVAE hyperparameters search space. See an official implementation[3]

## D  DATASETS

We used the following datasets:

- Abalone (OpenML)

- Adult (income estimation, Kohavi (1996))

- Buddy (Kaggle)

- California Housing (real estate data, Kelley Pace & Barry (1997))

- Cardiovascular Disease dataset (Kaggle)

- Churn Modeling (Kaggle)

- Diabetes (OpenML)

- Facebook Comments (Singh et al. (2015))

- Gesture Phase Prediction (Madeo et al. (2013))

- Higgs (simulated physical particles, Baldi et al. (2014); we use the version with 98K samples available at the OpenML repository Vanschoren et al. (2014))

- House 16H (OpenML)

- Insurance (Kaggle)

- King (Kaggle)

- MiniBooNE (OpenML)

- Wilt (OpenML)

---

[3]https://github.com/Team-TUD/CTAB-GAN-Plus
[3]https://github.com/sdv-dev/CTGAN

# E  ENVIRONMENT AND RUNTIME

Experiments were conducted under Ubuntu 20.04 on a machine equipped with GeForce RTX 2080 Ti GPU and Intel(R) Core(TM) i7-7800X CPU @ 3.50GHz. We used Pytorch 10.1, CUDA 11.3, scikit-learn 1.1.2 and imbalanced-learn 0.9.1 (for SMOTE).

As for runtime of the proposed method, it depends on the dataset and hyperparameters. We provide 3 examples below. All three examples use $T = 1000$ and $batch\_size = 4096$. Note that hyperparameters tuning contains 50 runs and takes usually 8-10 hours.

| Dataset | input_dim | model_layers | train_steps | n_to_sample | train_time | sample_time |
|---------|-----------|--------------|-------------|-------------|------------|-------------|
| CH | 16 | [256,1024,1024, 1024,1024,512] | 30k | 26k | 670s | 6s |
| HI | 28 | [512,1024,1024, 1024,1024,512] | 30k | 502k | 502s | 430s |
| FB | 146 | [512,1024] | 30k | 1264k | 783s | 470s |

Table 18: Training and sampling time for TabDDPM.

