# OpenReview forum: "TabDDPM: Modelling Tabular Data with Diffusion Models"
_ICLR.cc/2023/Conference — Submitted to ICLR 2023_

### Official Review · Reviewer_8b5K · 2022-10-25

**Confidence:** 2
**Correctness:** 1
**Technical Novelty And Significance:** 3
**Empirical Novelty And Significance:** 2
**Recommendation:** 3

**Clarity, Quality, Novelty And Reproducibility:**

Clarity
-------
Several important details are lacking from the paper.  The paper is also unclear at times (e.g., for Table 4, why are multiple column entries highlighted in bold?) and inconsistent (TVAE and CTABGAN should be included for comparison in Table 5).

Both the abstract and the intro could also more clearly state what the goal of TabDDPM is, i.e., it is generate synthetic tabular data.  This point is vague until well into the paper.

Quality
--------
The quality of the paper has significant room for improvement.  Firstly, by including CTGAN (which is widely regarded as the SOTA GAN tabular data synthesizer) and higher performing versions of SMOTE as benchmark competitors.  Secondly, the model itself lacks the ability to capture correlations between categorical and numerical features; this should be directly addressed.

Novelty
---------
As far as the reviewer is aware, the use of diffusion models for tabular data synthesis is novel.

Reproducibility
------------------
The authors have uploaded all relevant code to an anonymous repository.

**Strength And Weaknesses:**

Strengths
------------
-The use of diffusion models for tabular data modeling is interesting

Weaknesses
---------------
-The proposed method suffers from one major flaw: "TabDDPM uses the multinomial diffusion to model the categorical and binary features, and the Gaussian diffusion to model the numerical ones." <- This ignores correlations between categorical and numerical variables.  Consider a categorical feature, "Name," and a numerical feature, "Name_length."  Your model could not learn, or even approximate, this deterministic relationship.  Corresponding synthesized data will also be incorrect.

-"Each categorical feature is handled by a separate forward diffusion process, i.e., the noise components for all features
are sampled independently." <- Same criticism as above

-The authors state the utility of their method for privacy.  But the two aforementioned deficiencies render synthetic data incorrect, e.g., synthetic data may disrespect even obvious correlations.

-Why is baseline SMOTE used?  Other variants, like SVM SMOTE, are reported to work much better.

-Only TVAE is compared to.  However, CTGAN should also be considered,
as it's source is open (it is the package as TVAE, the CTGAN package)
and is the main synthesizer (TVAE was developed as a competitor in the
CTGAN paper).

-The paper lacks a large amount of important details.  E.g.: 1) "Smote... we generalize it" <- How? (2) Details for utilized software, both implementation (SMOTE) and version (sklearn), are lacking from the paper.

-"we evaluate ML efficiency only with respect
to the CatBoost model (Prokhorenkova et al., 2018), which is the leading GBDT implementation
providing state-of-the-art performance on tabular tasks" <- Please say
"arguably is the leading...," as XGBoost is more widely used and also
provides SOTA GBDT performance.

-No discussion on the runtime (training and synthesis) of TabDDPM.

**Summary Of The Paper:**

The authors propose TabDDPM, a denoising diffusion model for tabular data modeling and synthesis.  Within this framework, categorical and binary features are modeled using multinomial diffusion, and, completely independently, numerical features are modeled using Gaussian diffusion.  The model is compared to several recent deep generative models on the task of "Train on fake, test on real (TFTR)" [1] across different datasets considering the average of five classification/regression algorithms and, separately, Catboost.

[1] Jordon, James, Jinsung Yoon, and Mihaela Van Der Schaar. "PATE-GAN: Generating synthetic data with differential privacy guarantees." International conference on learning representations. 2018.

**Summary Of The Review:**

The proposed method has critical design flaws.  The evaluation and comparison to SOTA tabular data synthesized also has significant room for improvement.  I recommend rejection.

---

> ### Author Response · Authors · 2022-11-15
> **Reponse to Reviewer 8b5K**
>
> Thanks for the review.
>
> > This ignores correlations between categorical and numerical variables.
>
>
> This is not correct since the model takes as input both numerical and categorical features (Figure 1) and, thus, models the joint distribution.
>
> As for the forward process, it always samples noise components independently for different features (e.g. diagonal covariance matrix in Gaussian diffusion). This is a common way to define forward diffusion process. However, the model still learns joint distribution for a backward process though.
>
>
> > Other variants, like SVM-SMOTE, are reported to work much better.
>
> It is not suitable for our task, since SVM-SMOTE samples points close to a decision boundary and, thus, does not cover the whole distribution. We use SMOTE just to generate synthetic data as a linear combination of two real samples.
>
> > However, CTGAN should also be considered, as it's source is open
>
> You can find a full comparison below (and in Appendix B).
>
> | method   | AB    |    AD |    BU | CA    |    CAR |    CH |    DE |    DI | FB    |    GE |    HI | HO    | IN   | KI    |    MI |    WI |
> |:---------|:-------|-------:|-------:|:-------|-------:|-------:|-------:|-------:|:-------|-------:|-------:|:-------|:-------|:-------|-------:|-------:|
> | CTGAN    | 0.4204 | 0.7893 | 0.8673 | 0.6865 | 0.73   | 0.7229 | 0.6986 | 0.4588 | 0.4433     | 0.3325 | 0.5753 | 0.4329 | 0.7453 | 0.7717 | 0.7833 | 0.7488 |
> | TVAE     | 0.4328 | 0.7810  | 0.8638 | 0.7518 | 0.7174 | 0.7317 | 0.6564 | 0.7136 | 0.6853 | 0.434  | 0.6378 | 0.4926 | 0.7842 | 0.8238 | 0.9125 | 0.5006 |
> | CTABGAN  | --     | 0.7831 | 0.8552 | --     | 0.7171 | 0.6875 | 0.6437 | 0.731  | --     | 0.3922 | 0.5748 | --     | --     | --     | 0.8892 | 0.906  |
> | CTABGAN+ | 0.4672 | 0.7724 | 0.8844 | 0.5247 | 0.7327 | 0.7024 | 0.6865 | 0.7339 | 0.5088 | 0.4055 | 0.6639 | 0.5040 | 0.7966 | 0.4438 | 0.8920  | 0.7983 |
> | SMOTE    | 0.5486 | 0.7912 | 0.8906 | 0.8397 | 0.7323 | 0.7432 | 0.6930  | 0.6835 | 0.8035 | 0.6579 | 0.7219 | 0.6625 | 0.8119 | 0.8416 | 0.9323 | 0.9127 |
> | TabDDPM | 0.5499 | 0.7951 | 0.9057 | 0.8362 | 0.7374 | 0.7548 | 0.6910  | 0.7398 | 0.7128 | 0.5967 | 0.7218 | 0.6766 | 0.8092 | 0.8331 | 0.9362 | 0.9045 |
> | Real     | 0.5562 | 0.8152 | 0.9063 | 0.8568 | 0.7379 | 0.7403 | 0.6880  | 0.7849 | 0.8371 | 0.6365 | 0.7238 | 0.6616 | 0.8137 | 0.9070 | 0.9342 | 0.8982 |
>
> > No discussion on the runtime (training and synthesis) of TabDDPM.
>
> It depends on the dataset and hyperparameters. You can see 3 examples below. We report runtime of a single run with tuned hyperparameters. The number of diffusion steps is equal to 1000. Experiments were conducted on NVIDIA RTX 2080Ti. Tuning process usually takes 8+ hours.
>
> | dataset     | input_dim | model_layers                 | train_steps | n_to_sample | train_time | sample_time |
> |-------------|-----------|------------------------------|-------------|-------------|------------|-------------|
> | CH       | 16        | [256,1024,1024, 1024,1024,512] | 30k         | 26k         | 670s       | 6s          |
> | HI | 28        | [512,1024,1024, 1024,1024,512] | 30k         | 502k        | 502s       | 430s        |
> | FB | 146       | [512,1024]                     | 30k         | 1264k       | 783s       | 470s        |

---

### Official Review · Reviewer_d279 · 2022-10-25

**Confidence:** 5
**Clarity, Quality, Novelty And Reproducibility:** Not only the paper is not novel but a…
**Correctness:** 1
**Technical Novelty And Significance:** 1
**Empirical Novelty And Significance:** 1
**Recommendation:** 1

**Strength And Weaknesses:**

The paper is fairly well structured, however, it includes a few wrong claims. For example, it describes tabular data as data that includes different feature types, however, this is not necessarily a definition of tabular data. The features of tabular data can be only from a single family. In another part, the authors stated "Unlike unstructured images or natural texts, tabular data is typically structured," which is totally wrong. We have structure in image and NLP and the CNN is using this feature as a biased into the architecture. On the other hand, as stated in many tabular data papers, including SubTab [NeurIPS 2021] and VIME [NeurIPS 2020], and etc, the main problem of tabular data is that it does not have structure or the structured is not obvious as other domains.

Apart from that, the proposed method is not novel and the experiments are not convincing.

The number of features is too small in all experiments. The experiments are not showing how the generated samples are diverse.

One can also normalize features and then use diffusion models. This is not included in the baselines.

**Summary Of The Paper:**

The authors propose to combine multimodal and gaussian diffusion models as a new diffusion model that can handle small datasets with mixed kinds of features.

**Summary Of The Review:**

The paper combines two available diffusion models but the usefulness of the method for tabular data is wrong.

---

> ### Author Response · Authors · 2022-11-15
> **Response to Reviewer d279**
>
> Thanks for the review.
>
> >  In another part, the authors stated "Unlike unstructured images or natural texts, tabular data is typically structured," which is totally wrong. We have structure in image and NLP and the CNN is using this feature as a biased into the architecture.
>
> We respectfully disagree. The term "structured" corresponds to tabular data:
> https://keras.io/examples/structured_data/structured_data_classification_from_scratch/
> https://www.amazon.com/Deep-Learning-Structured-Data-Mark/dp/1617296724
>
> > The number of features is too small in all experiments
>
> We took datasets that were widely used in tabular data papers (`[1, 2, 3]`). And we think that 50 features is large enough.
>
> > The experiments are not showing how the generated samples are diverse.
>
> We follow computer vision techniques`[4]` and measure *density* and *coverage*, that are an improved versions of precision and recall, respectively. We report an average rank across all datasets, i.e. lower is better (1 is the best, 5 is the worst). Lower rank indicates higher density or coverage. The exact numbers for all datasets can be found in Appendix B.
>
> |          | density| coverage |
> |:---------|--------|---------:|
> | CTGAN    | 4.73   | 4.1      |
> | TVAE     | 3      | 4.0      |
> | CTABGAN+ | 3 .93  | 3.3      |
> | SMOTE    | 1.07   | 2.03     |
> | TabDDPM  | 2.27   | 1.57     |
>
> > One can also normalize features and then use diffusion models. This is not included in the baselines.
>
> We do normalize features in the proposed method. It can be observed in Figure 1. It is also stated in Section 4.
>
>
> * `[1]`: Zilong Zhao, Aditya Kunar, Hiek Van der Scheer, Robert Birke, Lydia Y. Chen. CTAB-GAN: Effective Table Data Synthesizing. In ACML 2021.
>
> * `[2]`: Richard Nock, Mathieu Guillame-Bert. Generative Trees: Adversarial and Copycat. In ICML 2022.
>
> * `[3]`: Yury Gorishniy, Ivan Rubachev, Valentin Khrulkov, Artem Babenko. Revisiting Deep Learning Models for Tabular Data. In NeurIPS 2021.
>
> * `[4]`: Muhammad Ferjad Naeem, Seong Joon Oh, Youngjung Uh, Yunjey Choi, Jaejun Yoo. Reliable Fidelity and Diversity Metrics for Generative Models. In ICML 2020.

---

### Official Review · Reviewer_x66q · 2022-10-25

**Confidence:** 3
**Correctness:** 3
**Technical Novelty And Significance:** 2
**Empirical Novelty And Significance:** 3
**Recommendation:** 5

**Clarity, Quality, Novelty And Reproducibility:**

Clarity: 5/10,
Quality: 4/10,
Novelty: 4/10,
Reproducibility 7/10.

**Strength And Weaknesses:**

Strength:
(1) It is interesting to apply the diffusion model to other tasks except for computer vision.
(2) The experimental results look good. Several widely used datasets are used, and the TabDDPM achieves the best results for most datasets.

Weaknesses:
(1) I have two main concerns:
  A. Can those models in Table 3 and Table 4 (the SMOTE, CTABGAN, and CTABGAN+ models) represent the SOTA results on all datasets? Actually, SMOTE is published in 2002.
  B. The recent task tends to have a large data size (i.e., FB dataset in this paper), while TabDDPM does not perform well on this dataset.
(2) Minor:
  A. The bold numbers of the last column in Table 4 are weird (DI and WI).

**Summary Of The Paper:**

This paper focuses on tabular problems. It proposes a diffusion model called TabDDPM to model tabular data. The TabDDPM can be universally applied to any tabular dataset and handles any type of feature. The experimental results show that the TabDDPM outperforms several SOTA models, which is evaluated on several datasets.

**Summary Of The Review:**

Tabular problem is an important task, and it is quite interesting to use the diffusion model to model tabular data. While the compared models (e.g., SMOTE) are not strong enough. The improvement of TabDDPM is obvious in Figure 3 and Figure 4, but not obvious in Table 3 and Table 4.

---

> ### Author Response · Authors · 2022-11-15
> **Response to Reviewer x66q**
>
> Thanks for the review and good concern about large datasets.
>
> > Can those models in Table 3 and Table 4 (the SMOTE, CTABGAN, and CTABGAN+ models) represent the SOTA results on all datasets?
>
> CTABGAN and CTABGAN+ are currently the state-of-the-art GAN-based methods for synthetic tabular data generation. TVAE was taken as the only VAE-based alternative. SMOTE is just a *simple* non-parametric baseline. And we think that it is crucial that many tabular generation papers overlook SMOTE as a baseline. Not all of the presented datasets were widely used for tabular data generation but that is why we have considered multiple models. We have also added CTGAN in Appendix B.
>
> > The recent task tends to have a large data size (i.e., FB dataset in this paper), while TabDDPM does not perform well on this dataset
>
> This is a good concern. We've performed additional experiments on two big datasets: Covertype (300k, 54 features, 7 classes), Credit (180k, 30 features, binclass). You are right that the gap between Real and TabDDPM increases with the size of dataset, but we think that performance is still good enough compared with the other baselines.
>
> |        | Covertype (F1) | Credit (F1) |
> |--------|-----------|--------|
> |CTGAN   | 0.471     |  0.500 |
> |TVAE    | 0.499     |  0.500 |
> |CTBGAN+ | 0.470     |  0.867 |
> |TabDDPM | 0.820     |  0.887 |
> |Real    | 0.939     |  0.918 |

---

### Official Review · Reviewer_qTDQ · 2022-10-25

**Confidence:** 3
**Correctness:** 3
**Technical Novelty And Significance:** 2
**Empirical Novelty And Significance:** 2
**Recommendation:** 3

**Clarity, Quality, Novelty And Reproducibility:**

**Clarity**
The method is fairly cleanly described and presented. The relation to previous work on Gaussian and multinomial diffusion is clear, as well as the comparison to previous work based on VAEs and GANs.

**Originality**
I'm broadly familiar with the generative modeling literature, but less familiar with the specific problem of generative modeling for tabular data. The proposed method seems to be a novel combination and application of existing diffusion methods for continuous and discrete data. The comparison of previous methods to a strong baseline is also highlighted as novel, and I appreciate that this is included.

**Strength And Weaknesses:**

**Strengths**
I think the overall motivation for the paper is good. It is often the case that generative models are specialized to particular media, and can be unsuitable out-of-the-box for generic tabular data.

The experimental results mainly highlight the unreasonable effectiveness of SMOTE as a baseline, beating both the VAE and GAN approaches. This is a useful observation.

**Weaknesses**
While demonstrating the strong SMOTE baseline is useful, it also highlights that the proposed method only matches this baseline. I'm also not sure about the comparison to SMOTE on sensitive data, both in terms of the metric used, and whether fitting generative models on sensitive data is a useful idea. The latter is discussed below, but in terms of the DCR metric, surely this only captures one possible vector of attack for a potential adversary? That is, a comprehensive comparison of privacy preservation in this setting would go beyond a simple proximity check, and instead marginalize over a number of possible attack vectors?

For a stronger baseline beyond SMOTE, I would have thought the authors would try a model which factorizes the continuous and discrete (non-ordinal) features. That is, train one model on the continuous features (e.g. a standard diffusion model), and then another model for the discrete features conditioned on the continuous features (e.g. a conditional autoregressive model), or vice versa. This would exploit the state-of-the-art generative models for both modalities, and would provide a strong baseline for a model trained on all features jointly to beat. Moreover, the statement 'In contrast, larger DCR values indicate that the generative model can produce something “new” rather than just copies of real data.' seems far too strong in that (i) it argues that the data and model capacity are sufficient to recover something close to the 'true' distribution, and (ii) DCR is a sufficient metric to determine this.

I may be unfamiliar with common practice, but I'm not sure that I buy the arguments regarding sensitive data and privacy. In particular, if data is sensitive enough to not be shared, training classification/regression models etc. on that data is itself a tricky task, potentially introducing issues of bias, fairness, etc., not to mention generic failure cases. To me, adding the extra step of first fitting a generative model to the data, which is then used to create synthetic data for input to the rest of the pipeline, only exacerbates these potential issues. For example, even though diffusion models have seen impressive breakthroughs in image generation, using diffusion models to generate e.g. medical scans for the purpose of enhancing a disease-detection model seems highly questionable.

Finally, when judging the performance of the proposed method, Figure 2 seems to just visualize a selection of the marginal feature distributions. This seems lacking, in that (i) the evaluation is qualitative, and (ii) surely we're primarily interested in judging the model's capacity to model the joint dependencies in the data correctly?

**Summary Of The Paper:**

The paper proposes TabDDPM, a diffusion model for generic tabular data consisting of both continuous and discrete features. The paper compares TabDDPM to existing VAE and GAN methods, and against a simple SMOTE baseline.

**Summary Of The Review:**

While the overall motivation of the paper is good, I'm not sure the proposed method does enough to justify itself over the SMOTE baseline (although demonstrating the effectiveness of that baseline is worthwhile in and of itself). I find it hard to recommend acceptance as things stand.

---

> ### Author Response · Authors · 2022-11-15
> **Response to Reviewer qTDQ**
>
> Thanks for the review.
>
> > ... generative models on sensitive data is a useful idea.
>
> We point out that using the samples from generative models in the privacy-oriented setups is common among both academicians and practitioners.
>
> In particular, several recent papers that address this setup are:
> - PATE-GAN: Generating synthetic data with differential privacy guarantees, ICLR 2019
> - Generative Models for Effective ML on Private, Decentralized Datasets, ICLR 2020
> - P3GM: Private High-Dimensional Data Release via Privacy Preserving Phased Generative Model, ICDE 2021
>
> Among practitioners, samples from generative models are used in ML competitions:
> https://www.kaggle.com/competitions/tabular-playground-series-sep-2021/overview
> and in commercial products:
> https://cloudblogs.microsoft.com/opensource/2021/02/18/create-privacy-preserving-synthetic-data-for-machine-learning-with-smartnoise/
>
> Overall, we believe that the setup we address in our submission is justified and demanded in applications.
>
> > Additional metrics of privacy preservation over a number of possible attack vectors?
>
> We follow `[1]` and `[2]` and measure a success of a full black-box privacy attack. The aim of attack is to infer whether a record belongs to its original training data. The score is measured using ROCAUC (higher score indicates higher success of attack). In terms of this metric, TabDDPM outperforms the SMOTE baseline significantly.
>
> |         |    AB |    AD |    BU |    CA |    CAR |    CH |    DE |   DI |    GE |    HI |    HO |    IN |    KI |   MI |    WI |
> |:--------|------:|------:|------:|------:|------:|------:|------:|-----:|------:|------:|------:|------:|------:|-----:|------:|
> | TabDDPM | 0.505 | 0.511 | 0.569 | 0.516 | 0.506 | 0.721 | 0.497 | 0.51 | 0.533 | 0.527 | 0.546 | 0.868 | 0.517 | 0.5  | 0.516 |
> | SMOTE   | 0.967 | 0.619 | 0.71  | 0.986 | 0.721 | 0.891 | 0.679 | 0.61 | 0.864 | 0.999  | 0.826 | 0.712 | 0.748 | 0.99 | 0.954 |
>
> > Additional baseline that factorizes numerical and categorical features.
>
> The model proposed by the reviewer is hardly implementable during the rebuttal time window. Moreover, we consider it to be more sophisticated to serve as a baseline.
>
> > (i) the evaluation is qualitative
>
> We follow `[3]` and measure the Wasserstein distance between numerical features and the Jensen–Shannon divergence between categorical features. We also report an L2 distance between correlation matrices (Figure 3 from the paper but now quantitative results). The results are presented as an average rank across all datasets, i.e. lower is better (1 is the best, 5 is the worst). Lower rank indicates lower WD distance, JS divergence and L2 distance. The exact numbers can be found in Appendix B.
>
> |          |   WD | JS  | L2-corr|
> |:---------|------|-----|--------|
> | CTGAN    | 3.33 | 4.77| 3.47   |
> | TVAE     | 4.2  | 3.92| 4.4    |
> | CTABGAN+ | 3.87 | 2.54| 3.4    |
> | SMOTE    | 1.67 | 2.15| 2.0    |
> | TabDDPM  | 1.93 | 1.62| 1.73   |
>
>
> > (ii) surely we're primarily interested in judging the model's capacity to model the joint dependencies in the data correctly
>
> Note, that Machine Learning utility is the main method to evaluate  the model’s capacity to model joint distributions in tabular data synthesis.
>
> * `[1]`: Chen, D., Yu, N., Zhang, Y., and Fritz, M. Gan-leaks: A taxonomy of membership inference attacks against generative models. In CCS, 2020.
> * `[2]`: Jaehoon Lee, Jihyeon Hyeong, Jinsung Jeon, Noseong Park, Jihoon Cho. Invertible Tabular GANs: Killing Two Birds with One Stone for Tabular Data Synthesis. In NeurIPS 2021.
> * `[3]`: Zilong Zhao, Aditya Kunar, Hiek Van der Scheer, Robert Birke, Lydia Y. Chen. CTAB-GAN: Effective Table Data Synthesizing. In ACML 2021.

---

### Decision · Program_Chairs · 2023-01-20

**Decision:**

Reject

**Justification For Why Not Higher Score:**

All reviewers recommend rejection

**Justification For Why Not Lower Score:**

N/A

**Metareview: Summary, Strengths And Weaknesses:**

The paper proposes a diffusion model for tabular data consisting of both continuous and discrete features. While the use of diffusion models for tabular data modeling is interesting and somewhat novel, there are several issues identified by the reviewers, among which the most dominant ones are: 1) the writing is a bit confusing and can mislead reviewers/readers, and 2) the experimental results are not convincing (lacking strong baselines, performances, big datasets and ablations). All reviewers recommended rejection, and I agree the paper should be further improved accordingly and cannot be accepted at this time.